# Mo–Nb–Si–B Alloy: Synthesis, Composition, and Structure

**Dmitrii Andreev \*** , **Yurii Vdovin** , **Vladimir Yukhvid** **and Olga Golosova**

Merzhanov Institute of Structural Macrokinetics and Materials Science, Russian Academy of Sciences, Chernogolovka, 142432 Moscow, Russia; vdovin-us@ism.ac.ru (Y.V.); yukh@ism.ac.ru (V.Y.); golosova@ism.ac.ru (O.G.)
\* Correspondence: ade@ism.ac.ru

**Abstract:** Cast refractory alloys Mo–Nb–Si–B were prepared by centrifugal self-propagating high-temperature synthesis (SHS) from metallothermic mixtures containing $MoO_3$, $Nb_2O_5$, Al, Si, and B powders, and additive of $Al_2O_3$ as a temperature-moderating and chemically inert agent. Variation in the centrifugal acceleration and amount of the additive affected the composition and structure of cast Mo–Nb–Si–B alloys. In a wide range of values, the combustion temperature was found to exceed 3000 K, and the combustion products were obtained as two-layer ingots of target Mo–Nb–Si–B alloy (lower) and $Al_2O_3$ slag (upper).

**Keywords:** combustion; self-propagating high-temperature synthesis (SHS); Mo-based cast alloy





## 1. Introduction

Mo–Si alloys have high resistance to oxidation in air at temperatures of 1000–1650 °C; however, at the intermediate temperatures of 600–800 °C, they are prone to catastrophic oxidation [1,2]. The addition of B makes it possible to form a dense borosilicate glass that protects ceramics against oxidation. Mo–Si–B alloys prepared by heating and subsequent cooling were shown in [3] to represent Mo-based solid solution with $Mo_3Si$ and/or $Mo_5SiB_2$ inclusions. These alloys possess a far greater oxidation resistance than previously known molybdenum ones, but they are not as good as $Mo_5Si_3$–$Mo_3Si$–$Mo_5SiB_2$ alloys [4], and yet they contain a plastic α-Mo phase. As mentioned in [5], varying the volume fraction and morphology of the α-Mo phase in these Mo–$Mo_3Si$–$Mo_5SiB_2$ intermetallics allows it to achieve high values of fracture toughness and creep strength. Additional doping with Nb strengthens the molybdenum matrix but leads to no changes in the structural composition [6,7].

Mo–Si–B or Mo–Nb–Si–B alloys manufactured by the powder metallurgy method exhibit high heat resistance and high-temperature strength, thereby having the potential for turbo-engine applications. Because of the high melting points of Mo–Si-based alloys, multistage methods of powder metallurgy are favorable over melting ones. One of them includes the following stages: (1) mechanical activation in a vertical attritor, for 10 h, to completely dissolve Nb, Si, and B in the Mo matrix; (2) sintering at 1450 °C, and (3) hot isostatic pressing (HIP) at 1500 °C and under a pressure of 200 MPA [8]. An alternative route is the cost-effective, productive, and environment-friendly centrifugal self-propagating high-temperature synthesis (SHS) process, which provides the synthesis of Mo–Nb–Si–B alloy from a mixture consisting of a thermite composition $MoO_3$/$Nb_2O_5$/Al/Si/B and an elemental composition Mo/Nb/Si/B, such as a temperature-moderating additive [9]. In this work, we added $Al_2O_3$ instead of the costly and scarce Mo and Nb.

Thus, this work aimed at the preparation of Mo–$Mo_3Si$–$Mo_5SiB_2$ alloy by centrifugal SHS of $MoO_3$–$Nb_2O_5$–Al–Si–B powder mixtures containing $Al_2O_3$ as a temperature-moderating agent with special emphasis on optimizing process conditions.

---

## 2. Thermodynamic Calculation

Table 1 shows the composition of the green mixture and nominal weight percentage of elements (Mo, Si, B, and Nb) in a combustion product.

**Table 1.** Composition of the green mixture and nominal weight percentage of elements in a combustion product (wt %).

|  | Mo | Si | Nb | B | $MoO_3$ | $Nb_2O_5$ | Al |
|---|---|---|---|---|---|---|---|
| **Green mixture** | – | 1.5 | – | 0.5 | 68.9 | 2.4 | 26.7 |
| **Combustion product** | 92.5 | 3.0 | 3.4 | 1.0 | – | – | – |

Thermodynamic calculation (Figure 1a, where *P* is mass fraction of phases in combustion products), carried out using the software package TERMO 2.0, ISMAN, Chernogolovka, Russia [10], showed that the combustion of $MoO_3$/$Nb_2O_5$/Al/Si/B mixture occurs at high temperatures in the range of 2225–3500 K, at which condensed combustion products—alloy (Mo–Nb–Si–B) and slag ($Al_2O_3$)—are in a liquid phase state.

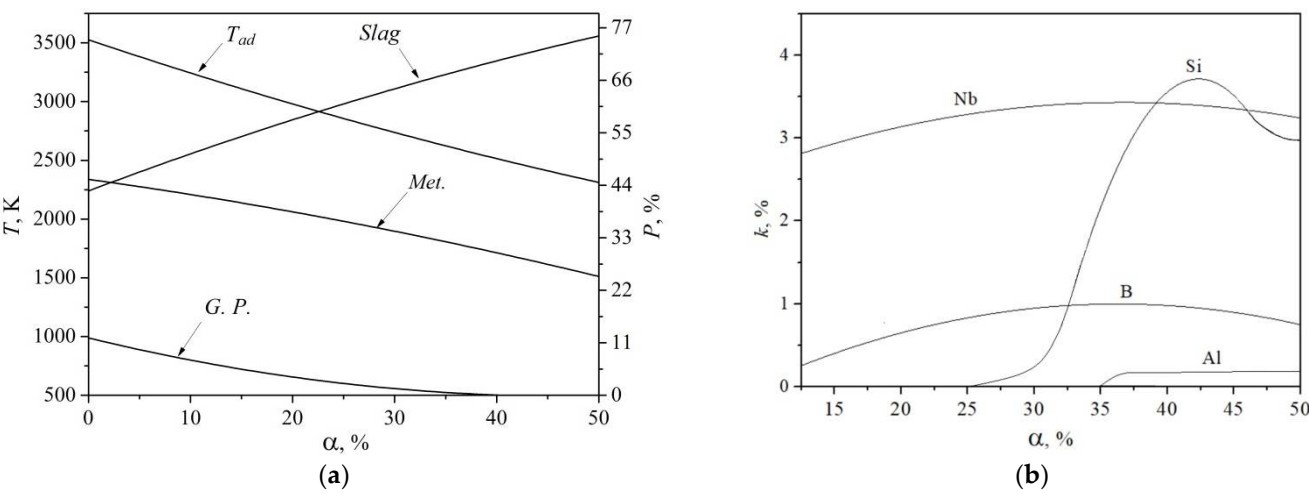

**Figure 1.** (**a**) Combustion temperature (*T*), mass fraction of phases in combustion products (*P*), and (**b**) percentage of constituents (the rest Mo) in metal phase (*k*) as a function of α. Here, $T_{ad}$ is the adiabatic temperature and G.P. is the gas phase.

Thermodynamic consideration predicts the formation of up to 10 wt % (*P*) of gaseous $Al_2O$, BO, $B_2O_2$, $SiO_2$, $NbO_2$, and so on (Figure 1a). The addition of aluminum oxide (α) to green composition decreases the combustion temperature *T* and mass fraction *P* of gas phase, as well as contributing to the appearance of B and Si in the combustion products (Figure 1b, where *k* is the percentage of constituents in the metal phase).

## 3. Experimental

Green mixtures containing powders of $MoO_3$, $Nb_2O_5$, Al, Si, and B in the amount of 40 g were ignited in a quartz tube (25 mm in diameter, 70 mm high) using a centrifugal machine described in [11] at the centrifugal acceleration *a* = 1–400 *g*.

In our experiments, the burning velocity U, the material loss ($\eta_1$) caused by sputtering, and the yield of target metallic phase into ingot $\eta_2$ were calculated using the following relationships:

$$U = h/t \tag{1}$$

$$\eta_1 = [(m_1 - m_2)/m_1] \times 100\% \tag{2}$$

$$\eta_2 = (m_{exp}/m_{cal}) \times 100\% \tag{3}$$

where h is the mixture height, t is the burning time, $m_1$ is the mass of green mixture, $m_2$ is the mass of combustion product, and $m_{exp}$ and $m_{cal}$ are the experimental and calculated mass of metallic ingot, respectively. $m_{cal}$ was worked out on the basis of chemical equation with the complete alumothermal reduction of initial oxides and alloying elements (Si and B) in the corresponding weighed portions.

The combustion products were characterized by scanning electron microscopy SEM (Carl Zeiss Ultra Plus microscope, Carl Zeiss, Jena, Germany) and X-ray diffraction analysis XRD (DRON-3M diffractometer, Cu-$K_\alpha$ radiation, Burevestnik, St. Petersburg, Russia). Concentrations of Mo, Nb, Si, and Al in the final product were determined by spectrophotometry. The determination of boron was carried out by potentiometric titration of mannitol-boric acid, while that of oxygen was performed by reductive melting in an inert carrier gas flow. The Vickers hardness of the synthesized samples was measured using a 100 g load and a 15 s loading time (Instron 402MVD tester, Wilson Instruments, Norwood, MA, USA).

## 4. Results

The experiments showed that the combustion of high-exothermic mixture is accompanied by the splashing of burning products. This was suppressed by the introduction of $Al_2O_3$ ($\alpha$) into the mixture and forces of artificial gravity. The mixtures were found to burn within the range $\alpha$ = 0–50%. As $\alpha$ increased within the indicated range, material loss $\eta_1$ significantly decreased, as seen in Figure 2. The cast product was formed for $\alpha$ = 0–40%. In this case, the combustion products were prepared as two-layer ingots: Mo–Nb–Si–B target alloy (lower) and $Al_2O_3$ slag (upper).

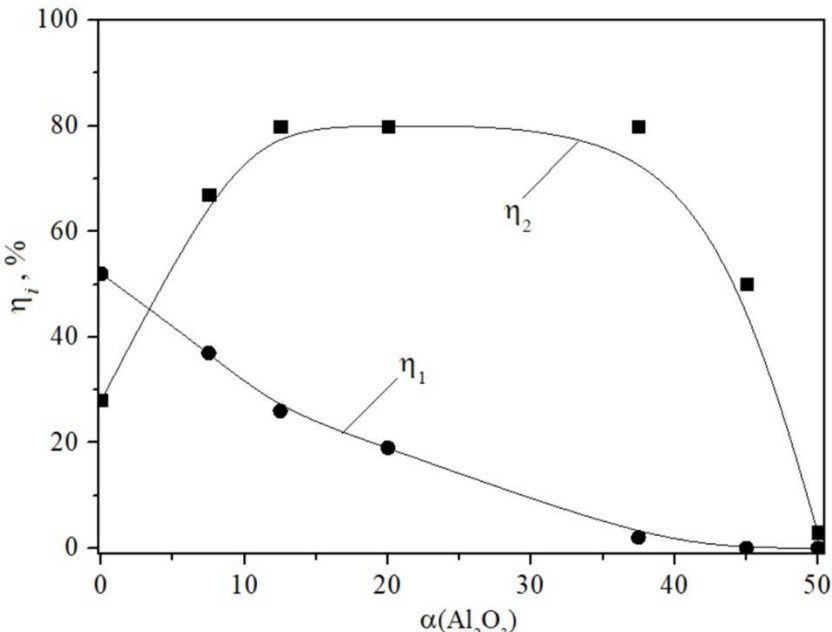

**Figure 2.** The values of $\eta_1$ и$\eta_2$ as a function of $\alpha$ ($m_1$ = 40 g, $a$ = 40 g).

As shown in Figure 3, the action of gravity forces makes it possible to increase the product yield $\eta_2$ from 80 to 90 wt %. An additional point to emphasize is that, as $a/g$ grows, the amount of pore space in the resultant cast Mo–Nb–Si–B material, according to microstructural analysis, decreases. For $a > 100$ g, a pore-free structure is formed.

Figures 4 and 5 illustrate the influence of $\alpha$ and $a/g$ on the percentage of constituents ($k$) of cast Mo–Nb–Si–B material, respectively. For $\alpha$ = 10–40% and $a$ = 40–400 g, EDS analysis for Nb gives 2–2.5 wt %, which is lower than the calculated value. The measured contents of Si and B show the values close to nominal ones (3–3.5 and 1 wt %, respectively).

With increasing $\alpha$ and $a/g$, Al concentration is seen to decrease from 6 to 2 wt %. The content of the main element (Mo) is within the range of 85–90 wt %.

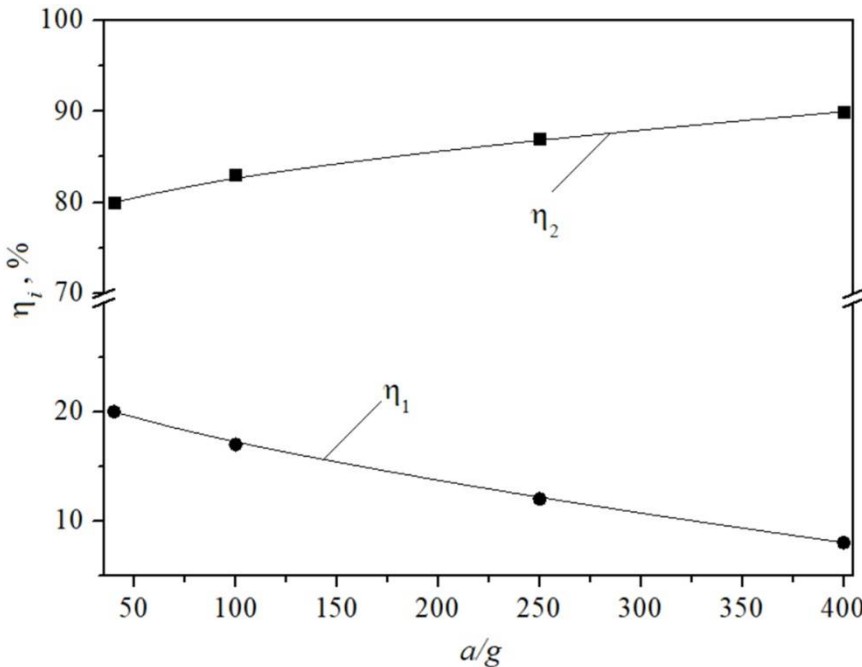

**Figure 3.** The values of $\eta_1$ and $\eta_2$ vs. $a/g$ ($m_1 = 40$ g, $\alpha = 20$ wt %).

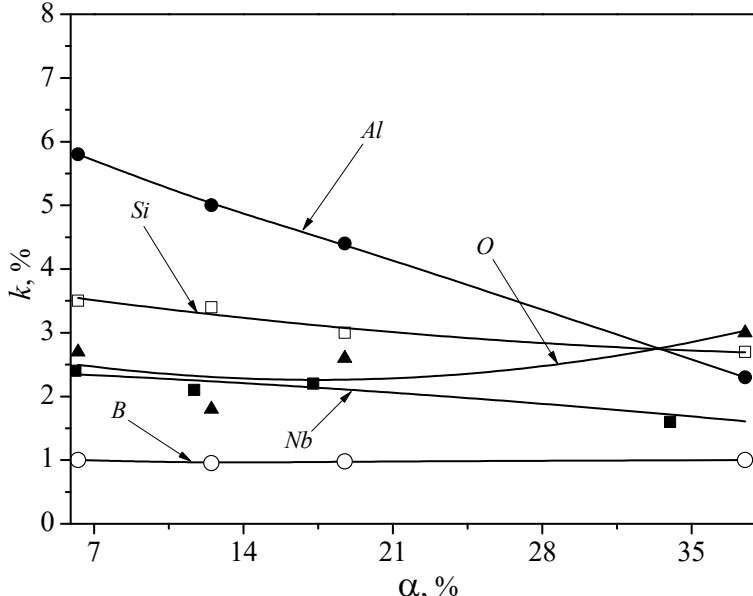

**Figure 4.** The percentage of constituents ($k$) in the resultant cast Mo–Nb–Si–B material as a function of $\alpha$ ($m_1 = 40$ g, $a = 40$ g).

The XRD pattern of Mo–Nb–Si–B ingot collects the peaks belonging to the following phases: (1) $\alpha$-Mo, (2) $Mo_3Si$, and (3) $Mo_5SiB_2$ (Figure 6a).

No other reacted phases were found. It is pertinent to note that XRD analysis shows no peaks of Nb insofar as it was completely dissolved in the Mo matrix phase during synthesis. A Mo/Si ratio in $Mo_3Si$ is 91.1/8.9 (wt %), while a Mo/Si/B ratio in $Mo_5SiB_2$ is 90.6/5.3/4.1. The slag is seen in Figure 6b to contain Mo in addition to conventional phase ($Al_2O_3$). Optic metallography confirmed the presence of individual spherical Mo particles in the oxide layer. Within the ranges $\alpha = 0$–10% and $\alpha = 40$–50%, there is a high

Mo content that favors the formation of a metal–ceramic structure in the slag layer. Thus, $\alpha = 10–40\%$ was chosen as optimal.

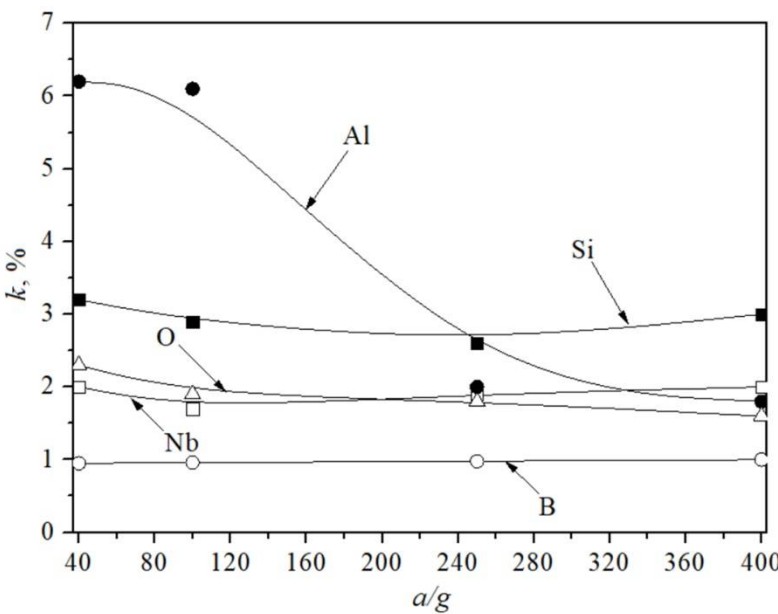

**Figure 5.** The percentage of constituents ($k$) in the resultant cast Mo–Nb–Si–B material as a function of $a/g$ ($m_1 = 40$ g, $\alpha = 20$ wt %).

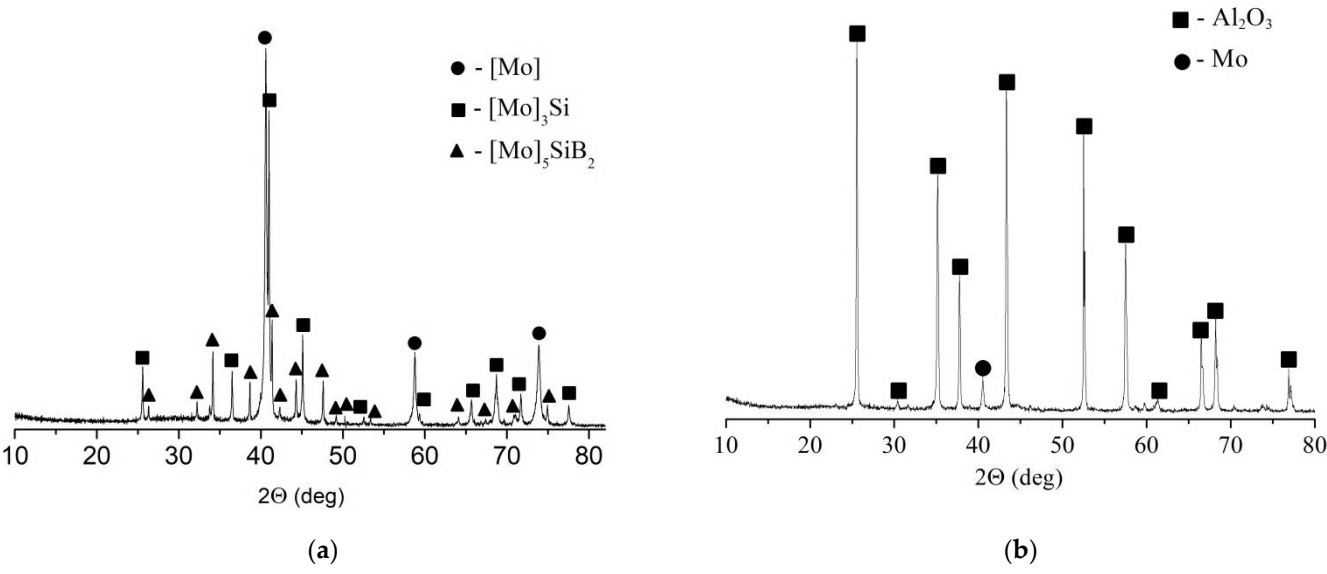

(**a**)                                        (**b**)

**Figure 6.** Diffraction patterns of (**a**) metallic phase and (**b**) slag derived from $a = 100$ g and $\alpha = 20$ wt %.

The SEM images presented in Figure 7 show a cast structure consisting of Mo solid solution (marked in Figure 7b by 1) and two intermetallic phases, $Mo_3Si$ and $Mo_5SiB_2$ (2 and 3, respectively). A quantitative analysis by X-ray diffraction revealed that the main phase is $Mo_3Si$; its volume fraction approximates 40%. $\alpha$-Mo phase has a volume fraction of around 30%, and, as seen in Figure 7a, forms to be discontinuous. According to [5], this fact can positively affect the creep strength of Mo–$Mo_3Si$–$Mo_5SiB_2$ intermetallics.

In order to evaluate the mechanical properties, we measured the Vickers hardness of ingots derived from $a = 100$ g and $\alpha = 20$ and 30 wt %. For $\alpha = 20$ wt %, the average hardness value was 1350 $HV$. It is 80 $HV$ lower than the hardness of ingot prepared at $\alpha = 30$ wt %.

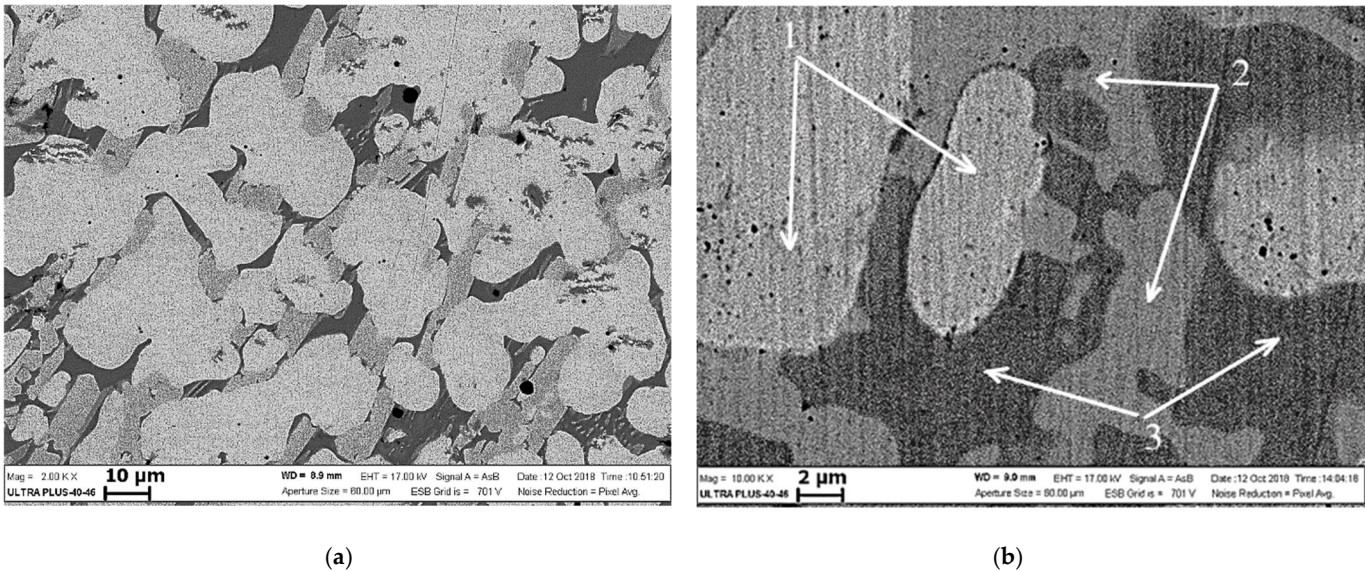

**Figure 7.** (**a**) Low- and (**b**) high-magnification SEM (scanning electron microscopy) images of cast alloys obtained at *a* = 100 *g* and *α* = 20 wt %. Magnification: (**a**) Mag 2.00 KX; (**b**) Mag 10.000 KX. Working distance (WD) = 9.9 mm. In (**b**): 1—Mo solid solution, 2—$Mo_3Si$, and 3—$Mo_5SiB_2$.

## 5. Discussion

The process of obtaining Mo-based cast alloys by centrifugal SHS includes 3 stages: (1) the combustion of $MoO_3/Nb_2O_5/Al/Si/B$ highly exothermic mixture and the formation of two-phase—Mo–Nb–Si–B and $Al_2O_3$—melt; (2) the gravitational separation of melts insoluble in each other under artificial gravity; and (3) the cooling, crystallization, and formation of the composition and structure of Mo–Nb–Si–B and $Al_2O_3$.

The combustion of $MoO_3/Nb_2O_5/Al/Si/B$ mixture is accompanied by the splashing of burning material. The latter is caused by the formation and release of gas under the action of Archimedean force. Thermodynamic calculation showed (Figure 1a) that up to 10 wt % of the gas phase (G.P.: $Al_2O$, BO, $B_2O_2$, $SiO_2$, $NbO_2$, and so on) can be formed. The introduction of $Al_2O_3$ into green mixture reduces gas formation, thereby markedly suppressing the splashing of mixture. However, as $Al_2O_3$ content increases, the combustion temperature decreases (see Figure 1a) and, as a result, the starting mixture loses its ability to burn.

The completion of the combustion results in a continuous $Al_2O_3$ melt containing Mo–Nb–Si–B drops. Under the action of gravity forces, heavy drops move to the bottom of the quartz mold and form a metal layer. The completeness of gravitational separation is determined by the ratio of velocities of the drops and the cooling of melt. Drop velocity is determined by the value of the centrifugal acceleration.

Under optimal conditions (*α* = 20–30% and *a* > 100 *g*), it is possible to suppress the splashing and to progress to 90 wt % of the yield of the target product (see Figure 3).

At the final stage of centrifugal SHS, the metal layer containing Mo–Nb–Si–B-based solid solution and two phases of $Mo_3Si$ and $Mo_5SiB_2$ are formed. SHS-produced Mo–$Mo_3Si$–$Mo_5SiB_2$ alloy is characterized by high hardness values, which far exceeds (by approximately 3 times) those attainable in Mo–Nb–Si–B alloy fabricated by a powder metallurgical method (425 *HV*) [7].

## 6. Conclusions

Mo-based composition materials reinforced with Nb, Si, and B, possessing good high-temperature and heat-resistance properties, can be prepared by centrifugal SHS from the highly exothermic composition $MoO_3/Nb_2O_5/Al/Si/B$ containing temperature-moderating additive ($Al_2O_3$) under the conditions of artificial gravity. Such materials seem

promising as a candidate for the next generation of high-temperature structural materials and as a high-hardness coating material [12].

**Author Contributions:** Conceptualization, D.A. and V.Y.; methodology, D.A., Y.V., and V.Y.; formal analysis, D.A., V.Y., and O.G.; investigation, D.A. and Y.V.; data curation, D.A. and O.G.; Writing—Original draft preparation, D.A.; Writing—Review and editing, D.A. and O.G.; visualization, Y.V. and O.G. All authors have read and agreed to the published version of the manuscript.

**Funding:** The study was not supported by any grant.

**Institutional Review Board Statement:** Not applicable.

**Informed Consent Statement:** Not applicable.

**Data Availability Statement:** Not applicable.

**Acknowledgments:** This research was performed by using the set of modern scientific instruments available for multiple accesses at the ISMAN Center of Shared Services.

**Conflicts of Interest:** The authors declare no conflict of interest.

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
