# Peer review of "Mo–Nb–Si–B Alloy: Synthesis, Composition, and Structure"

_metals, doi:10.3390/met11050803_

Round 1

Reviewer 1 Report

In this manuscript, the authors have systematically investigate the Mo-Nb-Si-B high temperature applicable materials using centrifugal SHS based on understanding high heat of mixing with the oxides. This seems quite unique to form novel phases using powder metallurgy.

Along this line, I am entirely agree to point out the issue to utilized the Mo-Nb-Si-B for several applications including high strength with elevating the temperature. Along this line, I would like to ask the authors to add some references at the introduction of this manuscript as similar to Mo-Si/Mo-Nb-Si etc to form the coating applications. 

Here is a references: 

Coatings, vol. 10 pp. 34 (2020) 

and so on.

This addition will be helpful to broaden the general understanding of Mo-Nb-Si-B alloy.

After this minor revision, I would like to recommend to publish it in this journal.

Author Response

Dear Reviewer,
We deeply appreciate the time and effort spent in reviewing our manuscript titled "Mo–Nb–Si–B Ceramics: Synthesis, Composition, and Structure". Your comments are all really valuable and very helpful for revising and improving our research.

Point 1: In this manuscript, the authors have systematically investigated the Mo-Nb-Si-B high temperature applicable materials using centrifugal SHS based on understanding high heat of mixing with the oxides. This seems quite unique to form novel phases using powder metallurgy.
Along this line, I am entirely agree to point out the issue to utilized the Mo-Nb-Si-B for several applications including high strength with elevating the temperature. Along this line, I would like to ask the authors to add some references at the introduction of this manuscript as similar to Mo-Si/Mo-Nb-Si etc to form the coating applications. Here is a references: Coatings, vol. 10 pp. 34 (2020) and so on. This addition will be helpful to broaden the general understanding of Mo-Nb-Si-B alloy.
After this minor revision, I would like to recommend to publish it in this journal.

Response 1: The suggested reference [12] was added in Conclusions.

Reviewer 2 Report

The authors in this contribution provide an interesting approach to the synthesis of Mo-Nb-Si-B alloys that is an attractive alternative to the costly and time consuming PM method. In addition to editorial work to correct the English expression the authors must improve the presentation of their calculations, results and figures as noted below.

  • First the use of the term ceramics to describe Mo-Nb-Si-B alloys is incorrect. The alloys are metallic and are not ceramics. The term ceramics should be replaced by alloy throughout the paper.
  • The terms in the calculations presented in figure 1 need to be clarified. What do Tad, G.P. and alpha represent?
  • In the caption in figure 2 m is given as 40g, but what does m mean? Is it m1, m2, mexp, or mcal.
  • In figures 4 and 5 the k values are given for all constituents except for Mo. Does the k for Mo represent the rest?
  • In figure 7 it is not clear how the individual phase identities in the microstructure were determined. The authors must offer evidence for the listed identities. Also, in figure 7 there are some dark spots that are not identified. Could these spots be pores or amorphous silica inclusions resulting from the synthesis reaction?

Author Response

Dear Reviewer,
We deeply appreciate the time and effort spent in reviewing our manuscript titled "Mo–Nb–Si–B Ceramics: Synthesis, Composition, and Structure". Your comments and term's correction are all really valuable and very helpful for revising and improving our research.

The authors in this contribution provide an interesting approach to the synthesis of Mo-NbSi-B alloys that is an attractive alternative to the costly and time consuming PM method. In addition to editorial work to correct the English expression the authors must improve the presentation of their calculations, results and figures as noted below.

Point 1: First the use of the term ceramics to describe Mo-Nb-Si-B alloys is incorrect. The alloys are metallic and are not ceramics. The term ceramics should be replaced by alloy throughout the paper.
Response 1: Corrected.

Point 2: The terms in the calculations presented in figure 1 need to be clarified. What do Tad, G.P. and alpha represent?
Response 2: Done. See p. 2, p. 5, and p. 9.

Point 3: In the caption in figure 2 m is given as 40g, but what does m mean? Is it m1, m2, mexp, or mcal.
Response 3: Corrected. See p. 10.

Point 4: In figures 4 and 5 the k values are given for all constituents except for Mo. Does the
k for Mo represent the rest?
Response 4: See p. 4.

Point 5: In figure 7 it is not clear how the individual phase identities in the microstructure were determined. The authors must offer evidence for the listed identities. Also, in figure 7
there are some dark spots that are not identified. Could these spots be pores or amorphous silica inclusions resulting from the synthesis reaction?
Response 5: The constituents of the microstructure were determined by combined use of EDS and XRD analysis. The dark spots in Figure 7 are pores.